# Association of Estimated Salt and Miso Intake with the Prevalence of Obesity in People with Type 2 Diabetes: A Cross-Sectional Study

**DOI:** 10.3390/nu13093014

**Published:** 2021-08-28

**Authors:** Fuyuko Takahashi, Yoshitaka Hashimoto, Ayumi Kaji, Ryosuke Sakai, Akane Miki, Yuka Kawate, Takuro Okamura, Noriyuki Kitagawa, Hiroshi Okada, Naoko Nakanishi, Saori Majima, Takafumi Senmaru, Emi Ushigome, Masahide Hamaguchi, Mai Asano, Masahiro Yamazaki, Michiaki Fukui

**Affiliations:** 1Department of Endocrinology and Metabolism, Graduate School of Medical Science, Kyoto Prefectural University of Medicine, Kyoto 602-8566, Japan; fuyuko-t@koto.kpu-m.ac.jp (F.T.); kaji-a@koto.kpu-m.ac.jp (A.K.); sakaryo@koto.kpu-m.ac.jp (R.S.); aknmk623@koto.kpu-m.ac.jp (A.M.); yukawate@koto.kpu-m.ac.jp (Y.K.); d04sm012@koto.kpu-m.ac.jp (T.O.); nori-kgw@koto.kpu-m.ac.jp (N.K.); conti@koto.kpu-m.ac.jp (H.O.); naoko-n@koto.kpu-m.ac.jp (N.N.); saori-m@koto.kpu-m.ac.jp (S.M.); semmarut@koto.kpu-m.ac.jp (T.S.); emis@koto.kpu-m.ac.jp (E.U.); mhama@koto.kpu-m.ac.jp (M.H.); maias@koto.kpu-m.ac.jp (M.A.); masahiro@koto.kpu-m.ac.jp (M.Y.); michiaki@koto.kpu-m.ac.jp (M.F.); 2Department of Diabetology, Kameoka Municipal Hospital, Kyoto 621-8585, Japan; 3Department of Diabetes and Endocrinology, Matsushita Memorial Hospital, Moriguchi 570-8540, Japan

**Keywords:** sodium excretion, sodium intake, fermented food, obesity

## Abstract

Salt intake is often estimated by the amount of sodium excreted in urine, and miso has been reported to increase it. This cross-sectional study investigated the relationship between obesity and high estimated salt intake with and without habitual miso consumption. Estimates of salt intake (g/day) were calculated using urinary sodium excretion, and a high estimated intake was defined as greater than the median amount of 9.5 g/day. Participants were divided into four groups based on estimated salt intake and miso consumption. Among 300 people, the proportions of obesity were 77.8% (*n* = 14/18), 40.2% (*n* = 53/132), 26.0% (*n* = 33/127), and 34.8% (*n* = 8/23) in the (+/−), (+/+), (−/+), and (−/−) groups of high estimated salt intake/habitual miso consumption, respectively. Compared with the (+/−) group, the adjusted odds ratios for obesity were 0.07 (95% confidence interval (CI): 0.02–0.26, *p* < 0.001), 0.16 (95% CI: 0.03–0.76, *p* = 0.022), and 0.14 (95% CI: 0.04–0.51, *p* = 0.003) in the (−/+), (−/−), and (+/+) groups, respectively. The presence of obesity was not much higher in people with high estimated salt intake with habitual miso consumption than that in people without. Clinicians should be aware that miso consumption promotes salt excretion, which may lead to an apparently higher estimated salt intake than actual.

## 1. Introduction

A high salt intake is known to elevate blood pressure and increase the risk of cardiovascular disease (CVD) [1,2]. For people with type 2 diabetes mellitus (T2DM), many guidelines recommend reducing salt consumption to prevent hypertension [3,4,5]. A previous study reported that high salt intake, estimated from a questionnaire, was related to an elevated incidence of CVD in people with T2DM [6]. Previous studies showed that salt intake, estimated from urinary excretion, is related to the risk of obesity [7,8,9,10,11].

Salt intake can be estimated by collecting urine over 24 hours (24 h urine), but, because this is difficult for both researchers and participants, it leads to incomplete data and low survey response rates; thus, spot urine is alternatively used to estimate salt intake [12,13,14], which is recommended by 2019 guidelines for the management of hypertension from the Japanese Society of Hypertension [15].

Miso (a fermented soybean food) is a traditional Japanese dish containing salt, vitamins, minerals, vegetable proteins, microorganisms, carbohydrates, and fat. Miso is known to inhibit angiotensin-converting enzyme (ACE) activity and cause diuresis, thus increasing sodium excretion in the kidneys and inhibiting sodium reabsorption in the renal tubules [16,17,18]. In a previous study, miso intake was shown to have a preventive effect on hypertension in non-hypertensive Japanese people, although miso itself increases dietary salt [19]. In addition, miso has been reported to suppress accumulation of visceral fat and the deposition of hepatic lipid in mice [19,20,21]. In fact, we recently revealed an association between miso and low body fat mass [22].

Therefore, we hypothesized that, because urinary estimates may not adequately assess the salt intake of people who consume miso habitually, a high level by itself might not be negative for obesity. To clarify this point, this cross-sectional study researched the association between estimated salt intake with and without habitual miso consumption and obesity in people with T2DM.

## 2. Method

### 2.1. Study Participants

This cross-sectional study used some data from the continuing KAMOGAWA-DM cohort study that began in 2014 [23]. It consists of outpatients from the Department of Endocrinology and Metabolism, Kyoto Prefectural University of Medicine Hospital (Kyoto, Japan) and the Department of Diabetes, Kameoka City Hospital (Kameoka, Japan). In this study, people who completed the questionnaire between January 2016 and December 2018 were included. Exclusion criteria were as follows: (1) no data on urinary sodium (Na) and creatinine (Cr), (2) incomplete questionnaire, (3) no data on multifrequency impedance body composition analysis, (4) abnormally high or low energy consumption (<600 or >4000 kcal/day) [24], (5) corticosteroid use [25], (6) absence of T2DM, or (7) the use of angiotensin-converting enzyme inhibitor (ACE) inhibitors or diuretics. This study obtained approval from the Local Research Ethics Committee (No. RBMR-E-466-6) and was carried out in accordance with the principles of the Declaration of Helsinki. All participating people submitted written informed consent.

### 2.2. Data Collection

Data on smoking, exercise, diabetes duration, and family history of diabetes were collected by means of a standardized questionnaire. Participants were divided into smokers and non-smokers, and regular exercisers were defined as people who played some kind of sport more than once a week. Blood pressure was measured automatically using an HEM-906 device (OMRON, Kyoto, Japan) in a sitting position in a quiet space after the participant had rested for 5 min. Additionally, data on medications, including insulin and antihypertensive drugs, were collected from the participants’ records. People with hypertension were defined as having a systolic blood pressure ≥ 140 mmHg, having a diastolic pressure ≥ 90 mmHg, or using antihypertensive drugs [26].

Furthermore, venous blood was obtained after overnight fasting, and the levels of fasting plasma glucose, high-density lipoprotein cholesterol, triglycerides, Cr, Na, potassium (K), and chloride (Cl) were measured. The estimated glomerular filtration rate (eGFR) was evaluated using the equation established by the Japanese Society of Nephrology, that is, eGFR = 194 × Cr^−1.094^ × age^−0.287^ or × 0.739 for women (mL/min/1.73 m^2^) [27]. Glycosylated hemoglobin (HbA1c) levels were estimated using high-performance liquid chromatography and expressed as National Glycohemoglobin Standardization Program units [28].

### 2.3. Definition of Obesity

Body weight (kg) and body fat mass (kg) were measured with a multifrequency impedance body composition analyzer, InBody 720 (InBody Japan, Tokyo, Japan) [29]. Body mass index (BMI, kg/m^2^) was derived by dividing body weight (kg) by the square of height (m). Ideal body weight (IBW) was calculated as follows: 22 multiplied by the square of the participant’s height in meters squared [30]. Percent body fat mass (%) was evaluated as follows: body fat mass (kg) divided by body weight (kg) × 100. Obesity was defined as body fat mass >30% for men and >35% for women [31].

### 2.4. Data on Habitual Dietary Intake, including Habitual Miso Consumption

A brief self-administered diet history questionnaire (BDHQ) was used to evaluate habitual food and nutrient intake during the preceding month based on the fifth edition of the Japanese Food Composition Table [32]. The data were automatically calculated according to the algorithm set by the developer. Participant data on energy, carbohydrate, protein, fat, miso, and alcohol consumption were calculated using the BDHQ. Consumption of energy (kcal/kg IBW/day), protein (g/kg IBW/day), fat (g/kg IBW/day), and carbohydrates (g/kg IBW/day) was calculated. An alcohol intake > 20 g/day was defined as a habitual [33]. Participants gave their daily miso soup consumption as none, <1 cup, or 1–8 cups or more. We defined non-habitual miso soup consumers as those who did not consume any [22].

### 2.5. Definition of Estimated Salt Intake

The participants provided urine samples from their second, early-morning urination episode. Urinalysis was carried out to measure concentrations of Na, K, and Cr. Urinary Na/K was computed by dividing urinary Na by urinary K. We estimated the 24 h salt intake using the Tanaka formula [13,14]:Estimated 24 h Cr (mg/day) = −2.04 × age + 14.89 × weight (kg) + 16.14 × height (cm) − 2244.5(1)
Estimated 24 h urinary Na (mEq/day) = 2/98 × urinary Na/10/Estimated 24 h Cr (mg/day)^0.392^
(2)
Estimated 24 h salt intake (g/day) = Estimated 24 h urinary Na (mEq/day)/17 (3)

People with high estimated salt intake were defined as those who consumed more than the median amount of 9.5 g/day.

### 2.6. Statistical Analyses

Participants were divided into two groups according to high or low estimated salt intake. They were then further divided into groups with or without habitual miso consumption. The four groups were as follows: low estimated salt intake without habitual miso consumption (−/−), high estimated salt intake without habitual miso consumption (+/−), low estimated salt intake with habitual miso consumption only (−/+), and high estimated salt intake with habitual miso consumption (+/+). Data were shown as means (standard deviation (SD)) or frequencies of potential confounding variables. Differences in continuous variables were evaluated using one-way analysis of variance (ANOVA) and the Tukey–Kramer test. Those in categorical variables were evaluated using the chi-square and the Holm tests.

We performed logistic regression analyses to investigate the relationship between obesity and the absence or presence of high estimated salt intake with or without habitual miso consumption. Independent variables were age, sex, exercise, smoking, alcohol, insulin, HbA1c, duration of diabetes, and energy intake.

Statistical analyses were conducted using EZR (Saitama Medical Center, Jichi Medical University, Saitama, Japan) [34], a graphical user interface for R (The R Foundation for Statistical Computing, Vienna, Austria). Differences with *p* < 0.05 were considered statistically significant.

## 3. Results

This study initially enrolled 523 people: 276 men and 247 women. Among these, 140 (64 men and 76 women) did not have T2DM; 8 (5 men and 3 women) did not undergo the bioelectrical impedance analysis test; 2 (1 man and 1 woman) had no urinary data; 22 (14 men and 8 women) had no BDHQ data; 6 (3 men and 3 women) had an extremely low or high daily energy intake; 10 (7 men and 3 women) were taking corticosteroids; 5 (4 men and 1 women) were using ACE inhibitors; and 30 (15 men and 15 women) were using diuretics. All were excluded from the study. Therefore, 300 (163 men and 137 women) participated in the study (Figure 1).

The clinical characteristics of the study participants are shown in Table 1. Mean age, BMI, and percent body fat mass were 65.7 ± 10.7 years, 24.5 ± 4.4 kg/m^2^, and 29.6 ± 9.1%, respectively. The proportion of obese participants was 36.0% (*n* = 108/300). The mean estimated salt intake was 9.4 ± 2.4 g/day. The proportion of participants with habitual miso consumption in the previous month was 86.3% (*n* = 259/300).

Table 2 shows the clinical characteristics of the participants based on the absence or presence of high estimated salt intake with or without miso consumption. The percentages of participants in the (−/−), (+/−), (−/+), and (+/+) groups were 7.7% (*n* = 23/300), 6.0% (*n* = 18/300), 42.3% (*n* = 127/300), and 44.0% (*n* = 132/300), respectively. The percentages of participants with obesity in the (−/−), (+/−), (−/+), and (+/+) groups were 34.8% (*n* = 8/23), 77.8% (*n* = 14/18), 26.0% (*n* = 33/127), and 40.2% (*n* = 53/132), respectively.

Furthermore, compared with the (+/−) group, the odds ratios (ORs) for obesity in the (−/−), (−/+), and (+/+) groups, were 0.16 (95% confidence interval (CI): 0.03–0.76, *p* = 0.022); 0.07 (95% CI: 0.02–0.26, *p* < 0.001); and 0.14 (95% CI: 0.04–0.51, *p* = 0.003) (Table 3), respectively.

## 4. Discussion

This study investigated the relationship between obesity and salt intake estimated from urine with or without habitual miso consumption in people with T2DM. We estimated overall sodium intake not only from salt because previous studies revealed that sodium intake itself was reported to be associated with a higher risk of CVD and obesity [1,2,6,7,8,9,10]. Even if we used estimated sodium intake instead of estimated salt intake, the result from urinary sodium excretion would have been the same.

Recent experimental studies showed that a high level of dietary salt promotes leptin production through adipocyte hypertrophy, causing increased white adipose tissue mass [35,36], and leptin resistance by activating the aldose reductase–fructokinase pathway in the liver and hypothalamus [8]. The increase in leptin secretion and corresponding decrease in tissue sensitivity to leptin leads to obesity [8,37,38]. In addition, salt intake induces obesity by activating the related diabetic gene’s N4-Acetylcytidine on the RNA level [39] and alternation of host gut microbiome [40,41,42].

High salt intake was not associated with all-causes mortality [43], but it was reported to lead to obesity [7,8,9,10]. As such, there are various reports on the effects of salt intake on health, which may be the result of a variety of factors, including miso. Interestingly, miso soup is known to have a diuretic effect on blood pressure [16]. Miso increases sodium excretion in the kidney and inhibits sodium reabsorption in the renal tubules. Nicotinamide extracted from soybeans inhibits ACE activity, and miso has the same effect [44,45]. Therefore, miso enhances sodium excretion, and people who consume it habitually might have been evaluated as having an excessive dietary salt intake.

The proportion of obesity in participants with high estimated salt intake was higher than that for those without high estimated salt intake (44.7% (*n* = 67/150) vs. 27.3% (*n* = 41/150), *p* = 0.003). Furthermore, high salt intake estimated from urine was related to obesity after adjusting for sex, age, insulin treatment, exercise, smoking, alcohol, diabetes duration, HbA1c, and energy intake (adjusted OR: 1.31 (95% CI: 1.10–1.56), *p* = 0.003), which is the same as the results of previous studies [5,7,8,42,43,44]. However, among those with a high urine-estimated salt intake, habitual miso consumption was not associated with obesity. 

There were certain limitations. First, the frequency of miso soup consumption data was self-reported, so the reliability of the data was uncertain. Second, a detailed dietary and physical activity survey would have been preferable for elucidating associations with estimated salt intake. Third, the estimated salt intake was not perfectly accurate. A previous study revealed that data from the Tanaka Formula underestimate 24 h values at low excretion levels and overestimate values at higher levels [46]. Salt intake estimated from excreted sodium is useful for assessing a population, but Tanaka’s formula is not suitable for individuals [14]. Estimated Na excretion, calculated from this formula, was reportedly correlated with measured 24 h Na excretion (*r* = 0.54, *p* < 0.01) [14]. Multiple, consecutive 24 h collections were required to evaluate regular salt intake accurately [47]. However, the error associated with using spot urine samples as a population is relatively low [48]. Fourth, we did not include plasma leptin levels. Thus, the causal relationship among estimated salt intake, leptin levels, and obesity was unclear. Fifth, the results of the questionnaire were based on the previous month’s consumption. Therefore, we cannot deny the possibility that the content of the food intake, when estimating the 24 h salt intake using the Tanaka formula, may have been different from the diet when the questionnaire was answered. Because of the cross-sectional nature of the study, it was not possible to show a causal relationship. Sixth, the sample size of this study, especially the two groups without habitual miso consumption, was relatively small. It would have been desirable to perform a Mendelian randomization analysis [49,50,51], but we did not have the genetic data. Further large sample studies are needed to clarify the validity of the results of this study.

## 5. Conclusions

Our study reported, for the first time, that obesity in people with high estimated salt intake and habitual miso consumption was lower than for people with high salt intake without miso, although the relationship between high salt intake and obesity has been recognized. Clinicians should be aware that estimated salt intake may be higher than actual salt intake in people with habitual miso consumption because miso consumption promotes salt excretion. 

## Figures and Tables

**Figure 1 nutrients-13-03014-f001:**
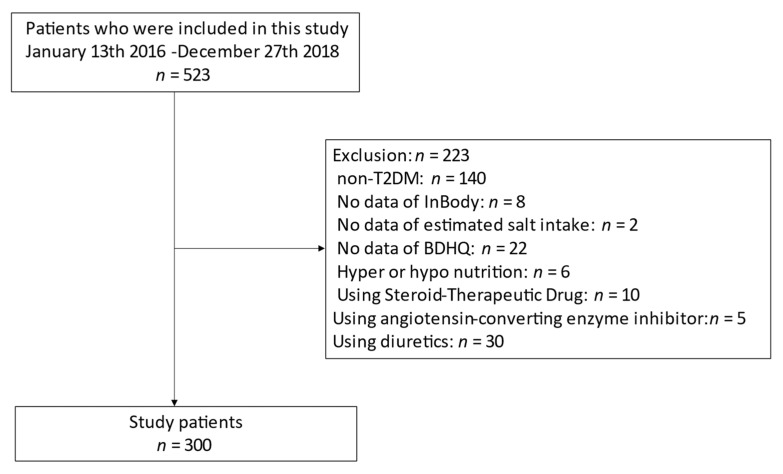
Inclusion and exclusion flow.

**Table 1 nutrients-13-03014-t001:** Clinical characteristics of study participants.

Characteristics	All *n* = 300
Sex (men/women)	163/137
Age (years)	65.7 (10.7)
Duration of diabetes (years)	13.7 (10.0)
Family history of diabetes (−/+)	160/140
Height (cm)	161.1 (9.4)
Body weight (kg)	63.5 (12.9)
BMI (kg/m^2^)	24.5 (4.4)
SBP (mmHg)	133.8 (18.8)
DBP (mmHg)	79.1 (11.1)
Antihypertensive drugs (−/+)	156/144
Presence of hypertension (−/+)	112/188
Insulin (−/+)	230/70
Habit of smoking (−/+)	255/45
Habit of exercise (−/+)	154/146
HbA1c (%)	7.4 (1.3)
HbA1c (mmol/mol)	57.3 (13.9)
Plasma glucose (mmol/L)	8.3 (2.8)
Triglycerides (mmol/L)	1.5 (0.9)
HDL cholesterol (mmol/L)	1.6 (0.4)
Creatinine (µmol/L)	70.0 (28.2)
eGFR (mL/min/1.73 m^2^)	72.0 (18.3)
Na (mmol/L)	139.6 (2.2)
K (mmol/L)	4.4 (0.4)
Cl (mmol/L)	103.2 (2.4)
Urinary Creatinine (µmol/L)	88.9 (56.9)
Urinary Na (mmol/L)	102.4 (47.6)
Urinary K (mmol/L)	46.6 (24.6)
Urinary Na/K (mmol/mmol)	2.7 (1.5)
Estimated salt intake (g/day)	9.4 (2.4)
High estimated salt intake (−/+)	150/150
Body fat mass (kg)	19.2 (8.6)
Percent body fat mass (%)	29.6 (9.1)
Presence of obesity (−/+)	192/108
Energy intake (kcal/IBW kg/day)	30.8 (9.8)
Protein intake (g/IBW kg/day)	1.3 (0.5)
Fat intake (g/IBW kg/day)	1.0 (0.4)
Carbohydrate intake (g/IBW kg/day)	3.9 (1.4)
Habitual miso consumption (−/+)	41/259
Alcohol consumption (g/day)	8.3 (20.7)
Habit of drinking alcohol (−/+)	262/38

Data are expressed as mean (standard deviation) or number. BMI, body mass index; SBP, systolic blood pressure; DBP, diastolic blood pressure; HbA1c, hemoglobin A1c; HDL, high-density lipoprotein; eGFR, estimated glomerular filtration rate; Na, sodium; K, potassium; Cl, chloride; IBW, ideal body weight.

**Table 2 nutrients-13-03014-t002:** Clinical characteristics of study participants according to the presence or absence of high estimated salt intake or habitual miso consumption.

High Estimated Salt Intake/Habitual Miso Consumption	(−/−)	(+/−)	(−/+)	(+/+)	*p*
	*n* = 23	*n* = 18	*n* = 127	*n* = 132	
Sex (men/women)	13/10	8/10	71/56	71/61	0.826
Age (years)	66.9 (14.7)	65.7 (9.9)	66.7 (10.4)	64.5 (10.3)	0.391
Duration of diabetes (years)	20.4 (12.1)	15.1 (8.1)	13.4 (9.8)	12.7 (9.6)	0.006
Family history of diabetes (−/+)	13/10	9/9	63/64	75/57	0.676
Height (cm)	158.1 (9.7)	159.4 (8.8)	161.2 (9.9)	161.7 (8.8)	0.336
Body weight (kg)	62.5 (12.1)	67.3 (12.8)	61.1 (12.6)	65.5 (13.0) ‡	0.026
BMI (kg/m^2^)	25.0 (4.6)	26.5 (4.8)	23.4 (3.8) †	25.1 (4.6) ‡	0.003
SBP (mmHg)	129.5 (16.3)	138.6 (21.4)	133.8 (19.4)	133.8 (18.4)	0.503
DBP (mmHg)	75.0 (10.5)	77.3 (9.1)	78.9 (11.2)	80.3 (11.2)	0.161
Antihypertensive drugs (−/+)	9/14	6/12	71/56	70/62	0.181
Presence of hypertension (−/+)	7/16	4/14	54/73	47/85	0.278
Insulin (−/+)	17/6	13/5	100/27	100/32	0.882
smoking (−/+)	19/4	16/2	107/20	113/19	0.938
exercise (−/+)	15/8	8/10	64/63	67/65	0.536
HbA1c (%)	7.8 (1.6)	7.9 (2.0)	7.3 (1.2)	7.3 (1.1)	0.111
HbA1c (mmol/mol)	61.5 (17.3)	63.2 (21.5)	56.7 (13.4)	56.4 (12.2)	0.111
Plasma glucose (mmol/L)	8.9 (2.9)	8.8 (4.5)	8.5 (3.0)	8.0 (2.2)	0.243
Triglycerides (mmol/L)	1.4 (0.8)	1.5 (0.7)	1.4 (0.9)	1.6 (1.0)	0.468
HDL cholesterol (mmol/L)	1.5 (0.5)	1.5 (0.3)	1.6 (0.4)	1.5 (0.5)	0.639
Creatinine (µmol/L)	73.1 (19.8)	66.9 (23.2)	73.6 (37.0)	66.3 (18.3)	0.183
eGFR (mL/min/1.73 m^2^)	67.7 (18.6)	74.3 (21.6)	69.3 (17.1)	75.0 (18.7)	0.048
Na (mmol/L)	139.5 (2.4)	139.7 (2.7)	139.4 (2.2)	139.8 (2.0)	0.434
K (mmol/L)	4.4 (0.4)	4.6 (0.5)	4.4 (0.4)	4.4 (0.4)	0.136
Cl (mmol/L)	103.0 (2.5)	102.7 (2.7)	102.9 (2.5)	103.5 (2.3)	0.148
Urinary Creatinine (µmol/L)	123.6 (52.6)	50.8 (31.5) *	117.6 (63.2) †	60.5 (30.7) *‡	<0.001
Urinary Na (mmol/L)	83.9 (37.5)	97.2 (45.5)	87.9 (39.6)	120.3 (50.6) *‡	<0.001
Urinary K (mmol/L)	58.8 (33.1)	30.3 (17.4) *	54.8 (25.7) †	38.8 (18.4) *‡	<0.001
Urinary Na/K (mmol/mmol)	1.7 (1.0)	3.6 (1.3) *	1.9 (1.0) †	3.5 (1.5) *‡	<0.001
Estimated salt intake (g/day)	7.0 (1.5)	11.3 (2.3) *	7.5 (1.4) †	11.3 (1.5) *‡	<0.001
Body fat mass (kg)	20.2 (8.6)	24.4 (9.6)	16.9 (7.1) †	20.5 (9.1) ‡	<0.001
Percent body fat mass (%)	31.8 (9.5)	35.6 (8.3)	27.2 (8.4) †	30.6 (9.1) ‡	<0.001
Presence of obesity (−/+)	15/8	4/14	94/33 †	79/53 †	<0.001
Energy intake (kcal/IBW kg/day)	25.4 (6.7)	27.9 (12.0)	31.6 (10.0) *	31.3 (9.5) *	0.019
Protein intake (g/IBW kg/day)	1.1 (0.3)	1.1 (0.4)	1.3 (0.5)	1.3 (0.5)	0.037
Fat intake (g/IBW kg/day)	0.9 (0.3)	1.0 (0.4)	1.0 (0.4)	1.0 (0.3)	0.467
Carbohydrate intake (g/IBW kg/day)	3.2 (0.9)	3.5 (2.1)	3.9 (1.3)	4.0 (1.4) *	0.038
Alcohol consumption (g/day)	1.7 (4.7)	3.5 (12.8)	9.8 (20.5)	8.5 (23.1)	0.260
Habit of drinking alcohol (−/+)	23/0	17/1	108/19	114/18	0.181

Data are expressed as mean (standard deviation), median (interquartile range), or number. The difference between group was evaluated by ANOVA and the Tukey–Kramer test for the continuous variables or chi-square test and the Holm test for the categorical variables. BMI, body mass index; SBP, systolic blood pressure; DBP, diastolic blood pressure; ACE inhibitors, angiotensin-converting enzyme inhibitors; HbA1c, hemoglobin A1c; HDL, high-density lipoprotein; eGFR, estimated glomerular filtration rate; Na, sodium; K, potassium; Cl, chloride; IBW, ideal body weight. *, *p* < 0.05 vs. (−/−); †, *p* < 0.05 vs. (+/−); and ‡, *p* < 0.05 vs. (−/+).

**Table 3 nutrients-13-03014-t003:** Odds ratio of the presence or absence of high estimated salt intake or habitual miso consumption for presence of obesity.

High Estimated Salt Intake/ Habitual Miso Consumption	Model 1		Model 2		Model 3	
	OR (95% CI)	*p*	OR (95% CI)	*p*	OR (95% CI)	*p*
(−/−)	0.15 (0.04–0.62)	0.009	0.14 (0.03–0.64)	0.011	0.16 (0.03–0.76)	0.022
(+/−)	ref	-	ref	-	ref	-
(−/+)	0.10 (0.03–0.33)	<0.001	0.09 (0.03–0.32)	<0.001	0.07 (0.02–0.26)	<0.001
(+/+)	0.19 (0.06–0.61)	0.005	0.17 (0.05–0.59)	0.005	0.14 (0.04–0.51)	0.003

Model 1 is unadjusted; model 2 is adjusted for sex, age, exercise, smoking, and alcohol; model 3 is adjusted for sex, age, exercise, smoking, alcohol, HbA1c, insulin, energy intake, and duration of diabetes.

## Data Availability

The datasets generated during and/or analyzed during the current study are available from the corresponding author on reasonable request.

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
