# Peer review of "Association of Estimated Salt and Miso Intake with the Prevalence of Obesity in People with Type 2 Diabetes: A Cross-Sectional Study"

_nutrients, 2021, doi:10.3390/nu13093014_

Round 1

Reviewer 1 Report

Nice paper, some comments below. Paper needs to be checked for grammar and sentence construction.

Lines 15-17: both sentences are missing a word. 

Line 22: some groups have small sample size; only 18 and 23 participants.

Line 60: word missing in sentence

Line 73: less/greater than signs missing. Is it <600 or >4000 kcal/d?

Line 86-88: references missing.

Line 110: what software was used to calculate energy and nutrient intakes?

Line 113: define 'regular consumption of miso'? Is it daily, or several times a day/week, etc.?.

Line 161: needs to be clarified that habitual miso consumption is the consumption in the previous month as that is what the BDHQ 

Line 242: Grammatical error; should be '..formula underestimates....'

Line 250: Sentence construction error. Also, cannot assess causality as this is a cross-sectional study.

Author Response

Nice paper, some comments below. Paper needs to be checked for grammar and sentence construction.

Response

We appreciate your comment. According to your comment, we have checked for grammar and sentence construction.

Point 1 Lines 15-17: both sentences are missing a word. 

Response

Thank you for your valuable comment. We have revised these sentences in the Abstract section as below.

Abstract (Line: 15-16)

“Salt intake is often estimated by the amount of sodium excreted in urine, and miso has been reported to increase it.”

Point 2 Line 22: some groups have small sample size; only 18 and 23 participants.

Response

Thank you for your comment. As you say, some groups, such as high estimated salt intake without high habitual miso consumption and low estimated salt intake without habitual miso consumption, have small sample size. Therefore, we have added this point as one of the limitations of this study in the Discussion section described as below.

Discussion (Line: 236-240)

“Sixth, the sample size of this study, especially the two groups without habitual miso consumption, was relatively small. It would have been desirable to perform a Mendelian randomization analysis [53-55], but we did not have the genetic data. Further large sample studies are needed to clarify the validity of the results of this study.”

Point 3 Line 60: word missing in sentence

Response

Thank you for your valuable comment. We have revised it as below.

Introduction (Line: 52-54)

“Therefore, we hypothesized that since urinary estimates may not adequately assess the salt intake of people who consume miso habitually, a high level by itself might not be negative for obesity.”

Point 4 Line 73: less/greater than signs missing. Is it <600 or >4000 kcal/d?

Response

Thank you for your comment. As you point out, the signs missing. This study has excluded energy intake <600 kcal/day or >4000 kcal/day. According to your comment, we have revised it in the Method section as below.

Method (Line: 66-67)

“Exclusion criteria were as follows: (omit) 4) abnormally high or low energy consumption (<600 or >4000 kcal/day) [24],”

Point 5 Line 86-88: references missing.

Response

Thank you for your comment. We have added the reference as below.

Reference

  1. Carretero, OA; Oparil, S. Essential hypertension. Part I: definition and etiology. Circulation 2000, 101: 329-335.

Point 6 Line 110: what software was used to calculate energy and nutrient intakes?

Response

Thank you for your comment. BDHQ is a self-administered questionnaire about diet in the past month, developed to estimate nutrients based on the fifth edition of the Japanese Food Composition Table. BDHQ automatically calculated the data of habitual food and nutrient intake according to the algorithm set by the developer. The validity of BDHQ were reported previously. Therefore, we have described this point in the Method section as below.

Method (Line: 99-102)

“A brief self-administered diet history questionnaire (BDHQ) was used to evaluate habitual food and nutrient intake during the preceding month based on the fifth edition of the Japanese Food Composition Table [32]. The data were automatically calculated according to the algorithm set by the developer.”

Point 7 Line 113: define 'regular consumption of miso'? Is it daily, or several times a day/week, etc.?.

Response

Thank you for your comment. Using BDHQ, the data of frequency of miso soup intake and the data of miso intake were obtained. Frequency of miso soup intake was asked how often they consume miso soup per day; none, less than 1 cup, 1 cup, 2 cups, 3 cups, 4 cups, 5 cups, 6 or 7 cups, or 8 cups or more per day. In this study, people without habitual miso intake were defined as those who did not consume miso soup at all in a day. According to your suggestion, we have added these points in the Methods section as below.

Method (Line: 106-108)

“Participants gave their daily miso soup consumption as none, <1 cup, or 1–8 cups or more. We defined non-habitual miso soup consumers as those who did not consume any [22].”

Point 8 Line 161: needs to be clarified that habitual miso consumption is the consumption in the previous month as that is what the BDHQ 

Response

Thank you for your valuable suggestion. As you say, the result of habitual miso consumption is gathered from the consumption diet of the previous month by BDHQ. According to your suggestion, we have added this point in the Result and Discussion sections as below.

Result (Line: 153-154)

“The proportion of participants with habitual miso consumption in the previous month was 86.3% (n = 259/300).”

Discussion (Line: 231-236)

“Fifth, the results of the questionnaire were based on the previous month’s consumption. Therefore, we cannot deny the possibility that the content of the food intake, when estimating the 24 h salt intake using the Tanaka formula, may have been different from the diet when the questionnaire was answered. Due to the cross-sectional nature of the study, it was not possible to show a causal relationship.”

Point 9 Line 242: Grammatical error; should be '..formula underestimates....'

Response

Thank you for your valuable comment. According to your comment, we have revised it in the Discussion section as below.

Discussion (Line: 223-224)

“A previous study revealed that data from the Tanaka Formula underestimate 24 h values at low excretion levels and overestimate values at higher levels [49].”

Point 10 Line 250: Sentence construction error. Also, cannot assess causality as this is a cross-sectional study.

Response

Thank you for your comment. As you say, this study cannot assess causality because this is a cross-sectional study. Thus, we have revised this sentence as below.

Discussion (Line: 230-232)

“Fourth, we did not include plasma leptin levels. Thus, the causal relationship among estimated salt intake, leptin levels and obesity was unclear.”

Reviewer 2 Report

Ln 35-36-  It is stated that reducing salt intake in many guidelines but there is only one citation included to support. These guidelines must be stated in short here to be more specific.

Ln 41. First sentence needs to be rewritten. Salt intake at the beginning and at the end of sentence makes it confusing.

Ln. 58-60: Check grammar

Ln 65: Delete Cohort study after the name of the study as you already mention, 'a prospective cohort study' soon after.

Section 2.1. First it is stated that it is a cohort study and then in the same paragraph it mentions it is a cross sectional study. Please check your study design carefully and report appropriately. Citation (22) does not match here. If this study was a part of large cohort study must be clearly specified. 

Section 2.2 State BP was measured in sitting position or supine position and if consistent for all participants.

Ln 87-88 needs citation.

Ln 94-96 needs citation.

Conclusion can be rewritten with few more details added to it, currently seems to be too short and language need some rework too.

Check reference 10

Overall, the study is interesting but needs rework in language at few places especially the Methods section. If possible get it edited by someone with expertise in language.

Thanks

Author Response

Point 1 Ln 35-36-  It is stated that reducing salt intake in many guidelines but there is only one citation included to support. These guidelines must be stated in short here to be more specific.

Response

Thank you for your valuable suggestion. According to your suggestion, we have added the references as below.

Reference

  1. American Diabetes Association. 5. Facilitating Behavior Change and Well-being to Improve Health Outcomes: Standards of Medical Care in Diabetes-2020. Diabetes Care 2020; 43(Suppl 1): S48-S65.
  2. Strom, BL; Anderson, CA; Ix, JH. Sodium reduction in populations: insights from the Institute of Medicine committee. JAMA 2013; 310: 31-32.

Point 2 Ln 41. First sentence needs to be rewritten. Salt intake at the beginning and at the end of sentence makes it confusing.

Response

Thank you for your suggestion. According to your suggestion, we have rewritten it in the Introduction section as below.

Introduction (Line: 38-42)

“Salt intake can be estimated by collecting urine over 24 hours (24 h urine) but because this is difficult for both researchers and participants, it leads to incomplete data and low survey response rates; thus, spot urine is alternatively used to estimate salt intake [12–14], which is recommended by 2019 guidelines for the management of hypertension from the Japanese Society of Hypertension [15].”

Point 3 Ln. 58-60: Check grammar

Response

Thank you for your comment. We have rewritten it in the Introduction section as below.

Introduction (Line: 52-44)

“Therefore, we hypothesized that since urinary estimates may not adequately assess the salt intake of people who consume miso habitually, a high level by itself might not be negative for obesity.”

Point 4 Ln 65: Delete Cohort study after the name of the study as you already mention, 'a prospective cohort study' soon after.

Response

Thank you for your comment. We have revised it in the Method section as below.”

Method (Line: 59-62)

“This cross-sectional study used some data from the continuing KAMOGAWA-DM cohort study that began in 2014 [23]. It consists of outpatients from the Department of Endocrinology and Metabolism, Kyoto Prefectural University of Medicine Hospital (Kyoto, Japan) and the Department of Diabetes, Kameoka City Hospital (Kameoka, Japan).”

Point 5 Section 2.1. First it is stated that it is a cohort study and then in the same paragraph it mentions it is a cross sectional study. Please check your study design carefully and report appropriately. Citation (22) does not match here. If this study was a part of large cohort study must be clearly specified. 

Response

Thank you for your comment. As you say, we should be clearly specified that this study was a part of large cohort study. According to your comment, we have revised this point in the Method section described as below.

Method (Line: 59-64)

“This cross-sectional study used some data from the continuing KAMOGAWA-DM cohort study that began in 2014 [23]. It consists of outpatients from the Department of Endocrinology and Metabolism, Kyoto Prefectural University of Medicine Hospital (Kyoto, Japan) and the Department of Diabetes, Kameoka City Hospital (Kameoka, Japan). In this study, people who completed the questionnaire between January 2016 and December 2018 were included.”

Point 6 Section 2.2 State BP was measured in sitting position or supine position and if consistent for all participants.

Response

Thank you for your comment. In this study, BP was measured in sitting position. Therefore, according to your comment, we have described in the Method section as below.

Method (Line: 76-78)

“Blood pressure was measured automatically using an HEM-906 device (OMRON, Kyoto, Japan) in a sitting position in a quiet space after the participant had rested for 5 min.”

Point 7 Ln 87-88 needs citation.

Response

Thank you for your comment. According to your comment, we have added the citation as below.

Reference

  1. Carretero, OA; Oparil, S. Essential hypertension. Part I: definition and etiology. Circulation 2000, 101: 329-335.

Point 8 Ln 94-96 needs citation.

Response

Thank you for your comment. According to your comment, we have added the citation as below.

Reference

  1. Antunes, MV; Wagner, SC; Camargo, JL; Linden, R. Standardization of method for determining glycosylated hemoglobin (Hb A1c) by cation exchange high performance liquid chromatography. Braz J Pharm Sci 2009; 45:651-657.

Point 9 Conclusion can be rewritten with few more details added to it, currently seems to be too short and language need some rework too.

Response

Thank you for your comment. According to your comment, we have revised Conclusion section as below.

“Our study reported, for the first time, that obesity in people with high estimated salt intake and habitual miso consumption was lower than for people with high salt intake without miso, although the relationship between high salt intake and obesity has been recognized. Clinicians should be aware that estimated salt intake may be higher than actual salt in-take in people with habitual miso consumption because miso consumption promotes salt excretion. ”

Point 10 Check reference 10

Response

Thank you for your suggestion. According to your suggestion, we have checked reference 10 and changed it as below.

  1. Ota, Y; Kitamura, M; Tsuji, K; Torigoe, K; Yamashita, A; Abe, S; Muta, K; Uramatsu, T; Obata, Y; Furutani, J; et al. Risk Reduction for End-Stage Renal Disease by Dietary Guidance Using the Gustatory Threshold Test for Salty Taste. Nutrients 2020; 12: 2703.

Point 11 Overall, the study is interesting but needs rework in language at few places especially the Methods section. If possible get it edited by someone with expertise in language.

Response

Thank you for your suggestion. According to your suggestion, this manuscript had edited by English expert.